# Clinical Outcomes Associated with SARS-CoV-2 Co-Infection with Rhinovirus and Adenovirus in Adults—A Retrospective Matched Cohort Study

**DOI:** 10.3390/ijerph20010646

**Published:** 2022-12-30

**Authors:** Quynh-Lam Tran, Gregorio Benitez, Fadi Shehadeh, Matthew Kaczynski, Eleftherios Mylonakis

**Affiliations:** 1Infectious Diseases Division, Warren Alpert Medical School of Brown University, Providence, RI 02903, USA; 2Geisel School of Medicine at Dartmouth, Hanover, NH 03755, USA; 3School of Electrical and Computer Engineering, National Technical University of Athens, 15780 Athens, Greece

**Keywords:** SARS-CoV-2, coronavirus disease 19, co-infection, rhinovirus, adenovirus, readmission, mortality, biomarker

## Abstract

(1) Background: Respiratory co-infections with severe acute respiratory syndrome coronavirus 2 (SARS-CoV-2) and other viruses are common, but data on clinical outcomes and laboratory biomarkers indicative of disease severity are limited. We aimed to compare clinical outcomes and laboratory biomarkers of patients with SARS-CoV-2 alone to those of patients with SARS-CoV-2 and either rhinovirus or adenovirus. (2) Methods: Hospitalized patients co-infected with SARS-CoV-2 and rhinovirus and patients co-infected with SARS-CoV-2 and adenovirus were matched to patients infected with SARS-CoV-2 alone. Outcomes of interest were the cumulative incidences of mechanical ventilation use, intensive care unit (ICU) admission, 30-day all-cause mortality, and 30-day all-cause readmission from the day of discharge. We also assessed differences in laboratory biomarkers from the day of specimen collection. (3) Results: Patients co-infected with SARS-CoV-2 and rhinovirus, compared with patients infected with SARS-CoV-2, had significantly greater 30-day all-cause mortality (8/23 (34.8%) vs. 8/69 (11.6%), *p* = 0.02). Additionally, median alanine transaminase (13 IU/L vs. 24 IU/L, *p* = 0.03), aspartate transaminase (25 IU/L vs. 36 IU/L, *p* = 0.04), and C-reactive protein (34.86 mg/L vs. 94.68 mg/L, *p* = 0.02) on day of specimen collection were significantly lower in patients co-infected with SARS-CoV-2 and rhinovirus in comparison to patients infected with SARS-CoV-2 alone. Clinical outcomes and laboratory markers did not differ significantly between patients with SARS-CoV-2 and adenovirus co-infection and patients with SARS-CoV-2 mono-infection. (4) Conclusion: SARS-CoV-2 and rhinovirus co-infection, compared with SARS-CoV-2 mono-infection alone, is positively associated with 30-day all-cause mortality among hospitalized patients. However, our lack of significant findings in our analysis of patients with SARS-CoV-2 and adenovirus co-infection may suggest that SARS-CoV-2 co-infections have variable significance, and further study is warranted.

## 1. Introduction

With the coronavirus disease 19 (COVID-19) pandemic in its second year, it has overlapped with other circulating pathogens [1,2,3,4]. Recent literature examining severe acute respiratory syndrome coronavirus 2 (SARS-CoV-2) co-infections with other respiratory pathogens have suggested that co-infection may be associated with more severe disease [5,6,7,8,9,10,11]. The mechanism by which co-infections enhance disease severity is poorly understood, but previous studies have proposed that it may be due to synergistic effects that promote immune activation and immunopathologies [12,13,14].

Research on how SARS-CoV-2 co-infection with non-influenza respiratory viruses affects disease courses is limited, as past studies have focused predominantly on co-infection with influenza virus [15,16,17,18,19,20,21,22,23]. Additionally, many studies on SARS-CoV-2 co-infections with other pathogens did not report outcomes [24,25,26]. As precautionary measures stringently enforced at the beginning of the pandemic loosen, we expect that there will be increased co-circulation of SARS-CoV-2 with other respiratory pathogens, which may have significant implications, particularly on vulnerable and immunocompromised individuals. Thus, a better understanding of SARS-CoV-2 co-infections is critical to anticipate disease burden and to evaluate the future of social distancing and other public health measures.

The objective of this retrospective matched cohort study was to examine clinical outcomes and laboratory biomarkers of hospitalized patients with SARS-CoV-2 co-infection with either human rhinovirus or adenovirus (hereafter referred to as SARS-CoV-2 + rhinovirus infection and SARS-CoV-2 + adenovirus infection, respectively) and of patients with SARS-CoV-2 alone.

## 2. Materials and Methods

### 2.1. Study Population and Setting

We utilized data from the electronic medical record system of the largest hospital system in Rhode Island, USA. The study population was comprised of patients aged 18 years and older who were admitted to the Rhode Island Hospital, The Miriam Hospital, or Newport Hospital between 1 December 2020 and 31 March 2022. As we focused primarily on SARS-CoV-2 co-infections with either rhinovirus or adenovirus, patients were included in the study if they had, from the same specimen collection, either SARS-CoV-2 + rhinovirus co-infection or SARS-CoV-2 + adenovirus co-infection that was confirmed by reverse transcription polymerase chain reaction. Specimen collection had to have occurred either during inpatient admission or in the emergency department prior to inpatient admission. We then matched these patients to control patients who had SARS-CoV-2 mono-infection, which was defined as testing positive for SARS-CoV-2 and negative for any other respiratory pathogen.

The study was conducted in accordance with the Declaration of Helsinki and approved by the Institutional Review Board (or Ethics Committee) of Lifespan (Lifespan IRB board #005120) for studies involving human participants. Patient consent was waived because the study is a retrospective chart review.

### 2.2. Matching Process

Hospitalized patients with SARS-CoV-2 + rhinovirus co-infection or SARS-CoV-2 + adenovirus co-infection were matched in a 1:3 ratio to patients with SARS-CoV-2 alone. If a patient with co-infection had more than three potential matched controls, then three matched controls were chosen at random. We matched based on age, sex, race and ethnicity, and time period of specimen collection. More specifically, when matching on age, we used the following predefined age groups: 0–9 years, 10–19, 20–29, 30–39, 40–49, 50–59, 60–69, 70–79, 80–89, 90 and above. Additionally, we matched on time period based on the following predefined time periods: (a) 1 December 2020–31 January 2021, (b) 1 February 2021–30 June 2021, (c) 1 July 2021–31 December 2021, and (d) 1 January 2022–31 March 2022. We utilized these predefined time periods for two main reasons: (1) to not limit our matching process to exact dates and thus risk an inefficient matching process, and (2) to encapsulate the evolving progression of the COVID-19 pandemic in Rhode Island in order to appropriately capture time-dependent contextual factors such as SARS-CoV-2 variant/sub-variant circulation and differential risk of severe clinical outcomes.

### 2.3. Clinical Outcome Assessment

Clinical outcomes of interest included the cumulative incidences of mechanical ventilation use during primary encounter, intensive care unit (ICU) admission during primary encounter, 30-day all-cause mortality, and 30-day all-cause readmission from the day of discharge from primary encounter. Only patients who were not admitted to the ICU or who had not received mechanical ventilation prior to or at the time of specimen collection were evaluated for incident ICU admission and mechanical ventilation use, respectively. Only patients who were discharged alive were evaluated for incident 30-day all-cause readmission, and only the first readmission was documented for analysis.

### 2.4. Baseline Laboratory Biomarker Assessment

We compared laboratory biomarker values collected on the day of specimen collection between patients with SARS-CoV-2 + rhinovirus co-infection and SARS-CoV-2 + adenovirus co-infection and the respective matched patients with SARS-CoV-2 alone. Specifically, we assessed differences in median alanine aminotransferase (ALT), aspartate aminotransferase (AST), C-reactive protein (CRP), creatinine, D-dimer, total lymphocyte count (TLC), platelet count, and white blood cell (WBC) count.

### 2.5. Statistical Analysis

Percentages are provided for categorical variables, and medians with interquartile ranges (IQRs) are provided for continuous variables. Either Wilcoxon rank sum tests or Fisher’s exact tests were performed to compare demographic and clinical factors between co-infection and mono-infection groups [27].

For the main analyses, the association between co-infection status and each clinical outcome of interest was assessed using Fisher’s exact test. We then used Wilcoxon rank sum tests to assess differences in baseline laboratory biomarkers. Analyses were performed using Stata software, version 17.0 (StataCorp, College Station, TX, USA), and statistical significance was set at α = 0.05. Exact matching was performed using the MatchIt package [28] in R [29].

## 3. Results

### 3.1. Study Disposition

Between 1 December 2020 and 31 March 2022, a total of 49 patients who were infected with SARS-CoV-2 and either rhinovirus or adenovirus were admitted to our hospitals. Our final cohort of patients was finalized after we (1) excluded patients who did not test positive for SARS-CoV-2 and rhinovirus or adenovirus from the same specimen collection and (2) excluded patients aged less than 18 years (Figure 1). Of note, none of the randomly selected controls were re-used in this analysis.

### 3.2. Baseline Characteristics of Patients with SARS-CoV-2 + Rhinovirus Co-Infection

As detailed in Table 1, we identified 23 patients with SARS-CoV-2 + rhinovirus co-infection with a median age of 77 years (IQR: 52–86). The median age of matched patients with SARS-CoV-2 mono-infection was 75 years old (IQR: 53–81). Additionally, out of the patients with SARS-CoV-2 + rhinovirus co-infection, 13/23 (56.5%) patients were male, 9/23 (39.1%) were Hispanic or Latino, and 10/23 (43.5%) patients were admitted between July 2021 and December 2021. In terms of oxygen support for patients with SARS-CoV-2 + rhinovirus co-infection, 13/23 (56.5%) were on room air, 6/23 (26.1%) were on low-flow oxygen delivery systems, and 4/23 (17.4%) were on high-flow oxygen delivery systems. Number of Elixhauser comorbidities, COVID-19 vaccination status, and remdesivir and corticosteroid use were comparable between patients with SARS-CoV-2 + rhinovirus co-infections and patients with SARS-CoV-2 mono-infection.

### 3.3. Baseline Characteristics of Patients with SARS-CoV-2 + Adenovirus Co-Infection

9 patients with a median age of 50 years had SARS-CoV-2 + adenovirus co-infection (IQR: 48–76) (Table 2). The median age of matched patients with SARS-CoV-2 mono-infection was 58 years (IQR: 42–72). Of these patients, 7/9 (77.8%) were male and 4/9 (44.4%) were Non-Hispanic White, and 6/9 (66.7%), patients were admitted between December 2020 and January 2021. For patients with SARS-CoV-2 + adenovirus co-infection, 3/9 (33.3%) patients were on room air, 6/9 (66.7%) patients were on low-flow oxygen delivery systems, and 0/9 (0%) patients were on high-flow oxygen delivery systems. Number of Elixhauser comorbidities, COVID-19 vaccination status, and remdesivir and corticosteroid use were also comparable between patients with SARS-CoV-2 + adenovirus co-infections and matched patients with mono-infection.

### 3.4. Clinical Outcomes for Patients with SARS-CoV-2 + Rhinovirus Co-Infection

In terms of 30-day all-cause mortality, patients with SARS-CoV-2 + rhinovirus co-infection had a statistically greater likelihood of experiencing mortality within 30 days compared to patients with SARS-CoV-2 alone (8/23 (34.8%) vs. 8/69 (11.6%), *p* = 0.02) (Table 3).

The cumulative incidences of 30-day all-cause readmission, ICU admission, and mechanical ventilation use in patients with SARS-CoV-2 + rhinovirus were numerically greater, but not statistically significant, than those of patients with SARS-CoV-2 mono-infection. Patients with SARS-CoV-2 + rhinovirus co-infection had a slightly numerically greater cumulative incidence of 30-day all-cause readmission compared to patients with SARS-CoV-2 mono-infection (2/19 (10.5%) vs. 5/64 (7.8%), *p* = 0.66). In terms of ICU admission, 3/23 (13.0%) patients with SARS-CoV-2 + rhinovirus co-infection were admitted to the ICU, compared to 3/68 (4.4%) patients with SARS-CoV-2 mono-infection. Regarding mechanical ventilation use, 2/23 (8.7%) patients with SARS-CoV-2 + rhinovirus co-infection required mechanical ventilation, compared to 2/69 (2.9%) patients with SARS-CoV-2 mono-infection (Table 3).

We also conducted an exploratory post hoc analysis to evaluate if the association between SARS-CoV-2 and rhinovirus co-infection is modified by baseline oxygen support status. Among patients on high-flow oxygen delivery systems, 4/4 (100%) patients with SARS-CoV-2 + rhinovirus co-infection and 2/2 (100%) patients with SARS-CoV-2 alone experienced mortality within 30 days. Furthermore, patients with SARS-CoV-2 + rhinovirus on room air and low-flow oxygen delivery systems had a numerically greater risk of death, compared to those with SARS-CoV-2 mono-infection (2/13 (15.4%) vs. 2/41 (4.9%), *p* = 0.242; 2/6 (33.3%) vs. 4/26 (15.4%), *p* = 0.310) (Table 4).

### 3.5. Laboratory Biomarkers for Patients with SARS-CoV-2 + Rhinovirus Co-Infection

As shown in Table 5, patients co-infected with SARS-CoV-2 + rhinovirus had a significantly lower median ALT (13 IU/L vs. 24 IU/L, *p* = 0.03), AST (25 IU/L vs. 36 IU/L, *p* = 0.04), and CRP (34.86 mg/L vs. 94.68 mg/L, *p* = 0.02), compared to patients with SARS-CoV-2 alone.

Although no other differences in laboratory biomarkers were statistically significant between the two groups, we also found that median D-dimer level was numerically lower in patients with SARS-CoV-2 + rhinovirus co-infection compared to patients with SARS-CoV-2 alone (269 ng/mL vs. 384 ng/mL, *p* = 0.49).

### 3.6. Clinical Outcomes for Patients with SARS-CoV-2 + Adenovirus Co-Infection

Clinical outcomes regarding 30-day all-cause mortality, 30-day all-cause readmission, ICU admission, and mechanical ventilation use did not differ significantly between patients co-infected with SARS-CoV-2 + adenovirus and patients infected with SARS-CoV-2 alone (Table 6). In terms of 30-day all-cause mortality, we found that a numerically lower number of patients with SARS-CoV-2 + adenovirus co-infection experienced mortality within 30 days in comparison to those with SARS-CoV mono-infection (0/9 (0%) vs. 3/27 (11.1%), *p* = 0.56). Similarly, a lower number of patients co-infected with SARS-CoV-2 + adenovirus required readmission within 30 days from day of discharge (0/9 (0%) vs. 3/23 (13.0%), *p* = 0.54). In terms of ICU admission, among patients with SARS-CoV-2 + adenovirus co-infection, 2/9 (22.2%) patients were admitted into the ICU during primary encounter, compared with 3/27 (11.1%) patients with SARS-CoV-2 mono-infection. Furthermore, 1/9 (11.1%) patients co-infected with SARS-CoV-2 + adenovirus and 3/27 (11.1%) patients infected with SARS-CoV-2 alone required mechanical ventilation use.

### 3.7. Laboratory Biomarkers for Patients with SARS-CoV-2 + Adenovirus Co-Infection

We did not find significant differences in laboratory biomarkers between patients with SARS-CoV-2 + adenovirus co-infection and patients with SARS-CoV-2 mono-infection, though patients with SARS-CoV-2 + adenovirus co-infection had a numerically greater median ALT (38.5 IU/L vs. 28.5 IU/L, *p* = 0.33) (Table 7).

## 4. Discussion

Continued circulation of SARS-CoV-2 with other seasonal respiratory viruses has led to increasingly higher rates of co-infection cases [25,26]. SARS-CoV-2 co-infections with influenza, in particular, have been shown to be associated with increased odds of mechanical ventilation and death [13]. However, because research on SARS-CoV-2 co-infections has primarily focused on co-infections with influenza, little is known about how co-infections with other respiratory viruses contribute to disease [17].

This retrospective matched cohort study examined clinical outcomes and laboratory biomarkers between patients co-infected with SARS-CoV-2 and either rhinovirus or adenovirus and patients with SARS-CoV-2 alone. We found that being infected with SARS-CoV-2 + rhinovirus was significantly associated with 30-day all-cause mortality in comparison to being infected with SARS-CoV-2 alone. Although not statistically significant, a higher proportion of patients with SARS-CoV-2 + rhinovirus co-infection were also readmitted within 30 days, admitted to the ICU, and placed on mechanical ventilation, compared to patients with SARS-CoV-2 alone. Additionally, among patients with the greatest degree of respiratory distress, all patients with SARS-CoV-2 + rhinovirus and SARS-CoV-2 mono-infection, respectively, died within 30 days. Among patients with lower respiratory distress (room air and low-flow oxygen delivery system use), there was a numerically greater risk of death among patients with co-infection. Future studies should consider effect modifiers, such as baseline oxygen support, when evaluating clinical outcomes in patients with co-infections compared to those with mono-infections.

Our findings that SARS-CoV-2 + rhinovirus co-infection adversely affects mortality are congruent with prior research that suggested unfavorable outcomes in patients with general respiratory co-infections [10,12,18,30,31]. In contrast, Chekuri et al. previously reported that being co-infected with both SARS-CoV-2 and other respiratory infections, including rhinovirus, was associated with a non-statistically significant lower frequency of severe clinical outcomes, compared with being infected with only SARS-CoV-2. Our different outcomes may be ascribed to how Chekuri et al. compared patients with SARS-CoV-2 co-infection generally to patients with SARS-CoV-2 alone, while we matched patients with SARS-CoV-2 + rhinovirus, specifically, to patients with SARS-CoV-2 alone [32].

In terms of laboratory biomarkers, Chekuri et al. also reported that patients with SARS-CoV-2 co-infections had lower CRP, ferritin, and fibrinogen compared to patients with mono-infection [32]. In the present study, we similarly found that SARS-CoV-2 + rhinovirus co-infection, compared to SARS-CoV-2 mono-infection, was associated with lower laboratory biomarkers, specifically lower median ALT, AST, and CRP. These laboratory tests have been found to be raised in patients with more severe COVID-19 disease [33,34,35,36,37,38,39]. Further research is needed to investigate the discrepancy between 30-day mortality and laboratory biomarkers, but we hypothesize that respiratory symptoms may predominate early on in the disease course of patients with SARS-CoV-2 co-infections, which may necessitate hospital care before markers of systemic inflammation and organ involvement rise.

Furthermore, whether SARS-CoV-2 + adenovirus co-infection exacerbates disease burden is still unclear, as only few studies report cases of co-infection with these two pathogens [40,41]. Notably, Swets et al. found that, in 136 patients with SARS-CoV-2 + adenovirus co-infection, co-infection was positively associated with in-hospital mortality, compared to SARS-CoV-2 mono-infection [13]. In our analysis of concurrent SARS-CoV-2 and adenovirus infection, we did not find any statistically significant differences in the clinical outcomes and laboratory biomarkers between patients with SARS-CoV-2 + adenovirus co-infection and SARS-CoV-2 mono-infection. This variation between our findings and those of Swets et al. may be due to the smaller number of patients co-infected with SARS-CoV-2 + adenovirus in our analysis. Our cohort of patients with SARS-CoV-2 + adenovirus co-infection was also younger than that of Swets et al.; the median age in our study was 50 years, while the median age in their study was 71 years. Relatedly, age may also explain the positive association with 30-day all-cause mortality between patients with SARS-CoV-2 + rhinovirus co-infection and those with SARS-CoV-2 mono-infection since patients in our study with SARS-CoV-2 + rhinovirus co-infection had a median age of 77 years. Thus, it may be that the association between co-infection status and mortality may be modified by age, although larger studies are needed for further assessment.

Regarding study limitations, our data collection relied solely on analyzing electronic medical records, so mortality may have been missed if it was not documented in the patients’ electronic medical records. However, we expect for missed outcome ascertainment to occur evenly across groups. In addition, given the retrospective nature of the study, causation cannot be determined based on the associations described in the study. Moreover, because laboratory biomarker values were not collected prospectively, we were unable to include other potential biomarkers of COVID-19 disease progression, such as neutrophil count, total bilirubin, and interleukin-6, and we could only evaluate values recorded on the day of specimen collection, due to missing data. Finally, comorbidity burden, COVID-19 vaccination status, and treatment use were not included in the matching process, but our data suggest that distribution across co-infection and mono-infection groups is comparable and any numerical differences may be a consequence of our small sample sizes. Nevertheless, future larger studies should consider the effect of these variables, notably vaccination status by the number of booster doses, on the incidence of clinical outcomes in patients with SARS-CoV-2 co-infection.

## 5. Conclusions

This study demonstrated that SARS-CoV-2 + rhinovirus co-infection was significantly positively associated with 30-day mortality compared to SARS-CoV-2 infection alone among hospitalized patients. Furthermore, median biomarkers of inflammation (ALT, AST, and CRP) were significantly lower in patients with SARS-CoV-2 + rhinovirus co-infection in comparison to patients with SARS-CoV-2 mono-infection. As preventative measures against COVID-19 continue to be loosened, vigilant surveillance and widespread testing for co-infection are essential. Additionally, clarifying the mechanism of viral interference in SARS-CoV-2 co-infection, specifically, and whether respiratory viral co-infections increase the risk of morbidity and mortality in certain patient populations is critical. Future research is needed to ascertain whether both non-modifiable and modifiable factors may predispose patients to have severe COVID-19 in the context of co-infection, especially in more at-risk groups, such as immunocompromised, pediatric, and geriatric patients.

## Figures and Tables

**Figure 1 ijerph-20-00646-f001:**
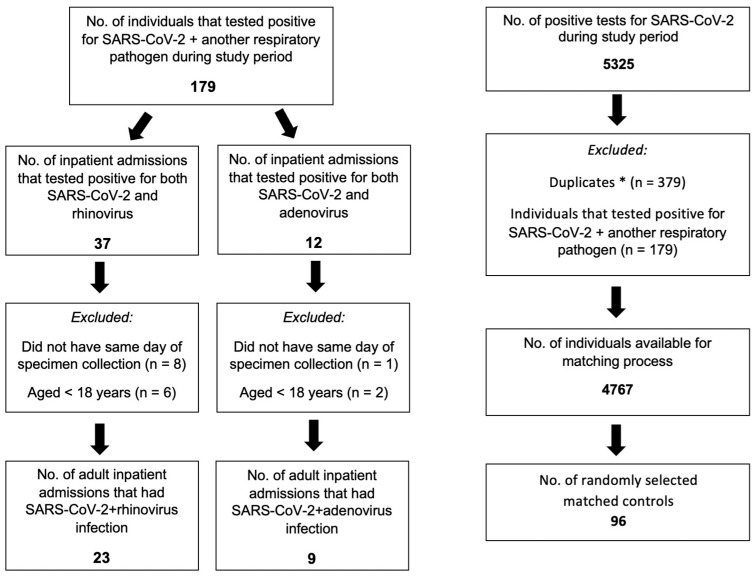
Patient Disposition Chart by Co-infection Status. *: If a patient tested positive more than once for SARS-CoV-2 during the study period, only the first instance of a SARS-CoV-2 positive test was eligible for selection in the analysis.

**Table 1 ijerph-20-00646-t001:** Matched cohort of adults with SARS-CoV-2 + rhinovirus infection or SAR-CoV-2 mono-infection.

Overall	SARS-CoV-2 + Rhinovirus *n* = 23	SARS-CoV-2 *n* = 69	*p*-Value
Age	77	75	0.71
[IQR]	[52–86]	[53–81]	
Patient Sex			-
Female	10 (43.5%)	30 (43.5%)	
Male	13 (56.5%)	39 (56.5%)	
Race/Ethnicity			-
Non-Hispanic White	8 (34.8%)	24 (34.8%)	
Hispanic or Latino	9 (39.1%)	27 (39.1%)	
Non-Hispanic Black	2 (8.7%)	6 (8.7%)	
Other/Unknown	4 (17.4%)	12 (17.4%)	
Time Period			-
December 2020–January 2021	3 (13.0%)	9 (13.0%)	
February 2021–June 2021	3 (13.0%)	9 (13.0%)	
July 2021–December 2021	10 (43.5%)	30 (43.5%)	
January 2022–March 2022	7 (30.4%)	21 (30.4%)	
Oxygen Support *			0.05
Room Air	13 (56.5%)	41 (59.4%)	
Low Flow	6 (26.1%)	26 (37.7%)	
High Flow	4 (17.4%)	2 (2.9%)	
No. of Elixhauser Comorbidities			0.219
0	2 (8.7%)	12 (17.4%)	
1–3	8 (34.8%)	27 (39.1%)	
4–6	6 (26.1%)	22 (31.9%)	
≥7	7 (30.4%)	8 (11.6%)	
COVID-19 Vaccination Status **			1.00
No	14 (60.9%)	43 (62.3%)	
Yes	9 (39.1%)	26 (37.7%)	
Remdesivir Use ***			0.231
No	14 (60.9%)	31 (44.9%)	
Yes	9 (39.1%)	38 (55.1%)	
Corticosteroid Use ***			0.608
No	6 (26.1%)	24 (34.8%)	
Yes	17 (73.9%)	45 (65.2%)	

*: Oxygen support at time of specimen collection was defined as room air (i.e., no support needed), use of low-flow delivery system, or high-flow delivery system (this included 2 patients with SARS-CoV-2 mono-infection and 1 patient with SARS-CoV-2 and rhinovirus co-infection on bilevel positive airway pressure). **: A patient was considered fully vaccinated against COVID-19 if the specimen collection date was ≥14 days after their second messenger RNA vaccine dose or ≥14 days after their Johnson & Johnson vaccine. All patients who did not meet this definition were not considered fully vaccinated. ***: Receipt of remdesivir or corticosteroid (i.e., dexamethasone or prednisone), respectively, during inpatient encounter. Percentages (%) may not sum to 100 due to rounding. Abbreviations: IQR: Interquartile Range, SARS-CoV-2: Severe Acute Respiratory Syndrome Coronavirus 2, COVID-19: Coronavirus Disease 2019.

**Table 2 ijerph-20-00646-t002:** Matched cohort of adults with SARS-CoV-2 + adenovirus infection or SAR-CoV-2 mono-infection.

Overall	SARS-CoV-2 + Adenovirus *n* = 9	SARS-CoV-2 *n* = 27	*p*-Value
Age	50	58	0.80
[IQR]	[48–76]	[42–72]	
Patient Sex			-
Female	2 (22.2%)	6 (22.2%)	
Male	7 (77.8%)	21 (77.8%)	
Race/Ethnicity			-
Non-Hispanic White	4 (44.4%)	12 (44.4%)	
Hispanic or Latino	3 (33.3%)	9 (33.3%)	
Non-Hispanic Black	1 (11.1%)	3 (11.1%)	
Other/Unknown	1 (11.1%)	3 (11.1%)	
Time Period			-
December 2020–January 2021	6 (66.7%)	18 (66.7%)	
February 2021–June 2021	1 (11.1%)	3 (11.1%)	
July 2021–December 2021	1 (11.1%)	3 (11.1%)	
January 2022–March 2022	1 (11.1%)	3 (11.1%)	
Oxygen Support *			0.146
Room Air	3 (33.3%)	17 (63.0)	
Low Flow	6 (66.7%)	10 (37.0%)	
High Flow	0 (0%)	0 (0%)	
No. of Elixhauser Comorbidities			0.946
0	3 (33.3%)	6 (22.2%)	
1–3	3 (33.3%)	11 (40.7%)	
4–6	2 (22.2%)	6 (22.2%)	
≥7	1 (11.1%)	4 (14.8%)	
COVID-19 Vaccination Status **			0.558
No	9 (100%)	24 (88.9%)	
Yes	0 (0%)	3 (11.1%)	
Remdesivir Use ***			0.252
No	2 (22.2%)	13 (48.2%)	
Yes	7 (77.8%)	14 (51.9%)	
Corticosteroid Use ***			0.223
No	1 (11.1%)	10 (37.0%)	
Yes	8 (88.9%)	17 (63.0%)	

*: Oxygen support at time of specimen collection was defined as room air (i.e., no support needed), use of low-flow delivery system, or high-flow delivery system. **: A patient was considered fully vaccinated against COVID-19 if the specimen collection date was ≥14 days after their second messenger RNA vaccine dose or ≥14 days after their Johnson & Johnson vaccine. All patients who did not meet this definition were not considered fully vaccinated. ***: Receipt of remdesivir or corticosteroid (i.e., dexamethasone or prednisone), respectively, during inpatient encounter. Percentages (%) may not sum to 100 due to rounding. Abbreviations: IQR: Interquartile Range, SARS-CoV-2: Severe Acute Respiratory Syndrome Coronavirus 2, COVID-19: Coronavirus Disease 2019.

**Table 3 ijerph-20-00646-t003:** Clinical outcomes between adults with SARS-CoV-2 + rhinovirus infection or SARS-CoV-2 mono-infection.

	Total *n* = 92	SARS-CoV-2 + Rhinovirus *n* = 23	SARS-CoV-2 *n* = 69	*p*-Value
30-day all-cause mortality				0.02
No	76 (82.6%)	15 (65.2%)	61 (88.4%)	
Yes	16 (17.4%)	8 (34.8%)	8 (11.6%)	
30-day all-cause readmission *	*n =* 83	*n =* 19	*n =* 64	0.66
No	76 (91.6%)	17 (89.5%)	59 (92.2%)	
Yes	7 (8.4%)	2 (10.5%)	5 (7.8%)	
ICU admission **	*n =* 91	*n =* 23	*n* = 68	0.17
No	85 (93.4%)	20 (87.0%)	65 (95.6%)	
Yes	6 (6.6%)	3 (13.0%)	3 (4.4%)	
Mechanical Ventilation				0.26
No	88 (95.7%)	21 (91.3%)	67 (97.1%)	
Yes	4 (4.3%)	2 (8.7%)	2 (2.9%)	

*: 30-day all-cause readmission was assessed among patients (*n* = 83) who were discharged alive. **: ICU admission was assessed among patients (*n* = 91) who were not in the ICU prior to or at time of specimen collection. One patient with SARS-CoV-2 mono-infection was excluded from ICU admission outcome assessment. Abbreviations: ICU: Intensive Care Unit, SARS-CoV-2: Severe Acute Respiratory Syndrome Coronavirus 2.

**Table 4 ijerph-20-00646-t004:** 30-day all-cause mortality between adults with SARS-CoV-2 + rhinovirus infection or SARS-CoV-2 mono-infection stratified by baseline oxygen support.

	SARS-CoV-2 + Rhinovirus	SARS-CoV-2	*p*-Value
Room Air			0.242
No	11 (84.6%)	39 (95.1%)	
Yes	2 (15.4%)	2 (4.9%)	
Low Flow			0.310
No	4 (66.7%)	22 (84.6%)	
Yes	2 (33.3%)	4 (15.4%)	
High Flow			-
No	0 (0%)	0 (0%)	
Yes	4 (100%)	2 (100%)	

Oxygen support at time of specimen collection was defined as room air (i.e., no support needed), use of low-flow delivery system, or high-flow delivery system (this included 2 patients with SARS-CoV-2 mono-infection and 1 patient with SARS-CoV-2 and rhinovirus co-infection on bilevel positive airway pressure). Percentages (%) may not sum to 100 due to rounding. Abbreviations: SARS-CoV-2: Severe Acute Respiratory Syndrome Coronavirus 2.

**Table 5 ijerph-20-00646-t005:** Laboratory biomarkers on day of specimen collection between patients with SARS-CoV-2 + rhinovirus infection or SARS-CoV-2 mono-infection *.

Overall	SARS-CoV-2 + Rhinovirus *n* = 23	SARS-CoV-2 *n* = 69	*p*-Value
ALT	*n* = 21	*n* = 58	0.03
Median (IU/L)	13	24	
[IQR]	[9–27]	[16–40]	
AST	*n* = 21	*n* = 58	0.04
Median (IU/L)	25	36	
[IQR]	[18–34]	[25–53]	
CRP	*n* = 16	*n* = 53	0.02
Median (mg/L)	34.86	94.68	
[IQR]	[11.62–87.42]	[43.25–142.02]	
Creatinine	*n* = 23	*n* = 66	0.99
Median (mg/dL)	0.94	0.90	
[IQR]	[0.60–1.47]	[0.74–1.27]	
D-dimer	*n* = 15	*n* = 50	0.49
Median (ng/mL)	269	384	
[IQR]	[180–721]	[240–592]	
TLC	*n* = 23	*n* = 66	0.16
Median (×10^9^/L)	0.9	0.8	
[IQR]	[0.5–1.7]	[0.5–1.2]	
Platelets	*n* = 23	*n* = 67	0.63
Median (×10^9^/L)	223	198	
[IQR]	[152–274]	[154–273]	
WBC	*n = 23*	*n* = 67	0.95
Median (×10^9^/L)	7.4	6.9	
[IQR]	[4.1–9.3]	[5.6–9.0]	

*: Differences in laboratory biomarkers on day of specimen collection were assessed among patients without missing data. Abbreviations: ALT: Alanine Aminotransferase, AST: Aspartate Aminotransferase, CRP: C-reactive protein, IQR: Interquartile Range, IU: International Unit, SARS-CoV-2: Severe Acute Respiratory Syndrome Coronavirus 2, TLC: Total Lymphocyte Count, WBC: White Blood Cell.

**Table 6 ijerph-20-00646-t006:** Clinical outcomes between adults with SARS-CoV-2 + adenovirus infection or SARS-CoV-2 mono-infection.

	Total *n* = 36	SARS-CoV-2 + Adenovirus *n* = 9	SARS-CoV-2 *n* = 27	*p*-Value
30-day all-cause mortality				0.56
No	33 (91.7%)	9 (100%)	24 (88.9%)	
Yes	3 (8.3%)	0 (0%)	3 (11.1%)	
30-day all-cause readmission *	*n* = 32	*n* = 9	*n* = 23	0.54
No	29 (90.6%)	9 (100%)	20 (87.0%)	
Yes	3 (9.4%)	0 (0%)	3 (13.0%)	
ICU admission				0.58
No	31 (86.1%)	7 (77.8%)	24 (88.9%)	
Yes	5 (13.9%)	2 (22.2%)	3 (11.1%)	
Mechanical Ventilation				1.00
No	32 (88.9%)	8 (88.9%)	24 (88.9%)	
Yes	4 (11.1%)	1 (11.1%)	3 (11.1%)	

*: 30-day all-cause readmission was assessed among patients (*n* = 32) who were discharged alive. Abbreviations: ICU: Intensive Care Unit, SARS-CoV-2: Severe Acute Respiratory Syndrome Coronavirus 2.

**Table 7 ijerph-20-00646-t007:** Laboratory biomarkers on day of specimen collection between patients with SARS-CoV-2 + adenovirus infection or SARS-CoV-2 mono-infection *.

Overall	SARS-CoV-2 + Adenovirus *n* = 9	SARS-CoV-2 *n* = 27	*p*-Value
ALT	*n* = 8	*n* = 26	0.33
Median (IU/L)	38.5	28.5	
[IQR]	[21–78.5]	[18–35]	
AST	*n* = 8	*n* = 26	0.97
Median (IU/L)	36	29.5	
[IQR]	[19–40]	[21–41]	
CRP	*n* = 7	*n* = 18	0.59
Median (mg/dL)	88.92	85.59	
[IQR]	[36.55–299.01]	[33.52–189.70]	
Creatinine	*n* = 9	*n* = 27	0.73
Median (mg/dL)	0.77	0.80	
[IQR]	[0.72–1.14]	[0.64–1.05]	
D-dimer	*n* = 8	*n* = 17	0.50
Median (ng/mL)	272.5	302	
[IQR]	[169.50–449]	[279–479]	
TLC	*n* = 9	*n* = 26	0.65
Median (×10^9^/L)	1.2	1.0	
[IQR]	[0.6–1.3]	[0.4–1.2]	
Platelets	*n* = 9	*n* = 27	0.38
Median (×10^9^/L)	181	206	
[IQR]	[132–223]	[150–258]	
WBC	*n* = 9	*n* = 27	0.16
Median (×10^9^/L)	7.2	6	
[IQR]	[6.9–8.3]	[5–8]	

*: Differences in laboratory biomarkers on day of specimen collection were assessed among patients without missing data. Abbreviations: ALT: Alanine Aminotransferase, AST: Aspartate Aminotransferase, CRP: C-reactive protein, IU: International Unit, IQR: Interquartile Range, SARS-CoV-2: Severe Acute Respiratory Syndrome Coronavirus 2, TLC: Total Lymphocyte Count, WBC: White Blood Cell.

## Data Availability

The datasets generated and/or analyzed during the current study are not publicly available due to HIPAA restrictions. De-identified summary data are available from the corresponding author upon reasonable request.

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
