# Peer review of "Clinical Outcomes Associated with SARS-CoV-2 Co-Infection with Rhinovirus and Adenovirus in Adults—A Retrospective Matched Cohort Study"

_ijerph, 2022, doi:10.3390/ijerph20010646_

Round 1

Reviewer 1 Report

Authors have done retrospective study of co-viral infection, which is in my view important basis for future studies.

Generally, it is very difficult to distinguish the clinical signs, disease severitiy, progression and prognosis, however, different laboratory parameters, molecular diagnostics, and other associated diagnostic modalitis like CT Scan, USG, MRI etc. can be used to characterize the specific organ involvement/severity grading and treatment selections.

As authors concluded, co-infection is positively associated and delay the recovery. There are many cases of mucormycosis coinfection association with SARS-CoV-2 have been reported. Therefore, it is important to study and have further investigation.

It is very important to note here that in the case of co-infection, generally we should need (practice) to have negative report of other potentially, probable co-infecting diseases say for an example, mucormycosis in case of  SARS-CoV-2 to have better, clear and absolute study.

As author targeted two viral co-infection, The laboratory parameters seems absolutely important in case of rhinovirus co-infection.

The study is well designed, executed and stadardized.

Result and Discussion is well. 

As it is retrospective study based on the data, authors have critically identified the limitation and presented well.

Article can be accepted.

Reviewer 2 Report

Dear Authors

Delighted to have the opportunity to review your manuscript, my comments are as follows

1.    I think it would be better to add some other data. For example data on vaccination, comorbidities (number of comorbidities), treatment received (especially corticosteroid), and the oxygen level at admission. Based on previous studies these variables could be confounders and I think including them is essential.

2.    Considering the matching process, did you exclude a (sars-cov-2 mono infection) patient once matched with a case of for example SARS-CoV-2 + rhinovirus from matching with SARS-CoV-2 + adenovirus or it was possible to include them again?

3.    In figure 1, what do you mean by "duplicates"?

4.    In the results section, please separate the results on baseline characteristics regarding rhinovirus and adenovirus, like the way you separated the outcome reports.

Reviewer 3 Report

Authors have done a very good work.

However following things should be addressed to improve it further:

-Authors have not mentioned the effect of vaccination in the COVID-19 patients co-infected with the rhino or adeno virus. Please mention that in the MS as well to clarify the significance of the current study.

-Similarly, a column for the included pateints' for their vaccination status might be essential.

-In the conclusion and most places of the MS authors have mentioned "SARS-CoV-2+rhinovirus co-infection". I suggest to add a space between the word SARS-CoV-2 and rhinovirus.
